# Effects of an Amphiphilic Micelle of Diblock Copolymer on Water Adsorption of Cement Paste

**DOI:** 10.3390/ma16062190

**Published:** 2023-03-09

**Authors:** Lei Dong, Fei Meng, Pan Feng, Qianping Ran, Chonggen Pan, Jianming He

**Affiliations:** 1School of Materials Science and Engineering, Southeast University, Nanjing 211189, China; 2State Key Laboratory of High-Performance Civil Engineering Materials, Jiangsu Sobute New Materials Co., Ltd., Nanjing 211103, China; 3School of Civil Engineering and Architecture, Ningbo Tech University, Ningbo 315100, China; 4Ningbo Construction Guangtian Component Co., Ltd., Ningbo 315100, China

**Keywords:** block copolymer, hydrophobicity, cement paste, compressive strength, water absorption

## Abstract

To reduce the inhibiting effects of polystyrene-based emulsion on the hydration process and strength development of cementitious materials, an amphiphilic diblock copolymer polystyrene-*block*-poly(acrylic acid) (PS-*b*-PAA) was synthesized via reversible addition-fragmentation chain transfer (RAFT) polymerization and demonstrated in cement paste system for improving the resistance to water absorption without significantly reducing 28-day compressive strength. Firstly, the dissolved PS-*b*-PAA was added into water, and it quickly self-assembled into amphiphilic 80 nm-sized micelles with hydrophobic polystyrene-based core and hydrophilic poly(acrylic acid)-based shell. The improved dispersion compared to that of polystyrene emulsion may minimize the inhibiting effects on strength development, as the effects of PS-*b*-PAA micelle as hydrophobic admixtures on rheological properties, compressive strength, water absorption, hydration process, and pore structure of 28-day cement pastes were subsequently investigated. In comparison with the control sample, the saturated water absorption amount of cement pastes with 0.4% PS-*b*-PAA was reduced by 20%, and the 28-day compressive strength was merely reduced by 2.5%. Besides, the significantly increased hydrophobicity instead of slightly decreased porosity of cement paste with PS-*b*-PAA may contribute more to the reduced water adsorption characteristics. The study based on prepared PS-*b*-PAA micelle suggested a promising alternative strategy for fabricating polystyrene-modified concrete with reduced water absorption and unaffected compressive strength.

## 1. Introduction

Cement-based materials, as typical porous materials, contain a series of different pore structures, including capillary pores, gel pores, and C-S-H interlayer pores [1], of which connected capillary pores are the key transmission channels of invasive substances, leading to the deterioration of structures including freeze-thaw, carbonization, chemical erosion, steel corrosion and so on. In this process, erosive ions adsorbed to the surface cement-based materials will gradually diffuse and penetrate the matrix and cause damage. Meanwhile, the capillary water absorption phenomenon in pores will drive rapid transportation and cause the enrichment of aggressive ions [2]. To reduce the water adsorption of cement-based materials, a series of approaches, including the densification of hydrated products, optimization of pore structure, and hydrophobic modification, are proposed.

In practical engineering applications, the reduced water adsorption of cement-based materials is generally achieved by tuning the water-binder ratio or using mineral admixtures. However, the reduced water-binder ratio may greatly affect the workability of fresh cement pastes, and the use of mineral admixtures such as silica fume or fly ash may reduce not only the early strength of hardened cement pastes but also increase the carbonization depth and the risk of shrinkage cracking [3,4,5]. Alternatively, much attention has been paid to the introduction of inorganic nanomaterials or hydrophobic organic components with specific advantages. Although inorganic nanomaterials such as nano-SiO_2_ [6,7], TiO_2_ [8], Fe_3_O_4_ [9], and carbon nanomaterials [10] can fill the connecting pores and reduce the harmful pores, the serious aggregation of nanomaterials in cement slurries and required high dosage amount may cause limited effects, increased shrinkage and poor workability [11,12]. 

In contrast, the addition of hydrophobic organic components seems more efficient as the pore interfaces inside the cement-based materials could be easily modified by small amounts of organic components, including silanes [13,14,15], polymers [16,17,18,19], long-chain fatty acid/salts [20,21,22] and so on. However, as these water-insoluble components are inclined to float and aggregate on the surface of cement slurry during the mixing, a heterogeneous matrix with uneven hydrophobic and mechanical properties will be formed [23]. In addition, these components are easily adsorbed to the surface of cement particles, causing an insoluble calcium soap adsorption layer. This layer may significantly inhibit the hydration process and cause problems of the slow development of concrete strength and insufficient strength in the later stage [23,24,25]. To address the above-mentioned problem, recently, these hydrophobic organic components are usually designed as emulsified particle forms, as to improve the compatibility and stability of components in water [26,27,28,29]. The most common polymer particle in aqueous dispersions is latex [30], such as dispersed butadiene-styrene copolymer mediated by the use of surfactants. The addition of polymer particles results in decreased porosity, enhanced workability, and decreased water adsorption amount [31,32]. Besides, the hydration could also be retarded by the presence of the polymer, causing lower compressive strength, modulus of elasticity, and hardness [31,33]. For instance, when butadiene styrene rubber (2.5–7.5% by weight of cement) was used as a cement modifier, improvements in properties contributing to the durability of concrete were discovered [30]. Liu investigated three types of polymer modifiers [18], including styrene-butadiene rubber latex, polyacrylic ester emulsion, and organic silicon waterproof agent (1–4% by weight of cement), and the incorporation of polymer modifier reduced the permeability of concrete and compressive strength. Despite the positive effects on impermeability, the preparation of emulsified polymer particles and control of the particle size both require massive efforts. And the emulsified particles may re-emulsify in humid alkaline conditions [29]. Besides, the applications based on dispersed polymer particles in core-shell structures are mostly limited to surface protection of cementitious materials [26,27,28], with barely any insights into the bulk water-repellent modification of cementitious materials.

In this paper, amphiphilic micelles with core-shell nanostructure are innovatively applied as an organic admixture to enhance the water-repellency of cement pastes. Polystyrene, which is commonly used as a material replacement to enhance the water absorption properties of concrete [34], is selected as a representative hydrophobic component to demonstrate the viability of design on materials. Owing to the potential control on molecular weight/composition of amphiphilic diblock copolymer (BCP) polystyrene-*block*-poly(acrylic acid) (PS-*b*-PAA) enabled by reversible addition-fragmentation chain-transfer polymerization (RAFT) [35], the micelle size and structure could be precisely tuned according to the specific demands. The satisfied compatibility with water is achieved by the self-assembled core-shell nanostructure of copolymers. The micro-morphology and micro-properties of micelle were first characterized comprehensively. Meanwhile, the effects of different PS-*b*-PAA dosages (0.1%, 0.2%, and 0.4%) on the hydrophobicity, water resistance, compressive strength, and hydration process of cement paste were investigated. The gradual increased PS-*b*-PAA dosage significantly contributed to the decreased hydrophilicity of cement paste. For 0.4% wt dosage, the added micelles effectively reduced the water adsorption amount of cement paste by 20.1% at 56 h and successfully maintained 28-day compressive strength in comparison with the control sample. Besides, the governing factors for the increased water resistance were further analyzed. Herein, the study contributes to new references for the potential application of amphiphilic micelles consisting of block copolymers for bulk water-repellent modification of cement-based materials.

## 2. Experimental

### 2.1. Materials and Methods

#### 2.1.1. Materials

Styrene purchased from Sigma-Aldrich was filtered through an alumina column to remove the inhibitors and distilled them before usage. Acrylic acid purchased from Shanghai Maclean Biochemical Technology Co., Ltd. (Shanghai, China) was distilled before usage. 4-Cyano-4-(thiobenzoylthio)pentanoic acid as RAFT agent was purchased from Sigma-Aldrich and used as received. Azobisisobutyronitrile (AIBN) as initiator was purchased from Sigma-Aldrich and recrystallized before usage. 1000 Da cut-off dialysis membranes were purchased from Feiyu Bio (Nantong, China) and rinsed in deionized water before usage. The other chemical reagents were analytical grade and applied without further purification. The chemical compositions of the PII 42.5 ordinary Portland cement (OPC) from Shi Jing cement company complying with GB 8076-2008 [36] are shown in Appendix A.

#### 2.1.2. Characterization

The chemical structures of synthesized copolymers were analyzed with an instrument (^1^H NMR, Avance III HD 600 MHz, Bruker) using chloroform-*d* as the solvent. Size exclusion chromatography analysis was performed on an instrument (SEC, PL-GPC220, Agilent), and the number-average molecular weights (*M*_n_) and molecular weight distributions (*M*_w_/*M*_n_) were determined by SEC with polymer/tetrahydrofuran solution at a flow rate of 1.0 mL/min at 40 °C and calibrated with polystyrene. The microstructure of the investigated PS-*b*-PAA micelle was conducted using transmission electron microscopy (TEM, Talos F200X, Thermo Fisher Scientific). The mineral compositions of cement paste were characterized by thermogravimetric analysis (TG, MDSC2910, American TG Company) and X-ray powder diffraction (XRD, SMARTLAB3KW, RIKEN Co., Ltd., Japan) with the range of 5–65° and the step size of 0.02°. The XRD identification of hydration products was performed by calculating Bragg’s relationship and automatically comparing it with Match software, making use of the PDF (Powder Diffraction File) database. For TG analysis, the CH content of the sample was estimated from Equation (1) [37].
(1)$$\mathrm{CH}\;\mathrm{content}=\frac{W_{400}-W_{500}}{W_{800}}\times \frac{m_{\mathrm{CH}}}{m_{H_{2}O}}+\frac{W_{500}-W_{800}}{W_{800}}\times \frac{m_{\mathrm{CH}}}{m_{{\mathrm{CaCO}}_{3}}}$$
where W denotes the recorded weight at specific temperatures; 400 and 500 present the weight loss temperatures of CH at 400 °C and 500 °C, respectively, while 500 and 800 are weight loss temperatures of CaCO_3_ at 500 °C and 800 °C, respectively; m_CH_, $m_{H_{2}O}$, and $m_{{\mathrm{CaCO}}_{3}}$ are the molecular mass of CH, H_2_O, and CaCO_3_, respectively.

The dispersion test was performed to reveal the distinctive dispersion states of PS-*b*-PAA and PS homo-polymer particles in water. The mean size and size distribution of PS-*b*-PAA micelle were determined by dynamic light scattering (DLS, CGS-3, ALV Co.) using a wide-angle light scattering instrument, which can monitor particles in the size range 1 nm–5 μm. The PS-*b*-PAA micelle solution was treated in an ultrasonic bath for 2 min and then diluted to 1 mg/g using deionized water for DLS measurements. The zeta potentials of the PS-*b*-PAA micelle in water were measured with a DT310 zeta potential analyzer. The cement paste sample (the weight ratio of water to cement is 0.38) with PS-*b*-PAA micelle (0.2 wt% of cement paste) was stirred (low speed for 2 min and high speed for 2 min), and the rheological properties were tested by R/SP-SST stress-controlled cement rheometer containing a helix type geometry (Brookfield, WI, USA). Both the yield stress and plastic viscosity results of the cement paste were calculated through the Bingham model. Compressive strength and water absorption tests were carried out to measure the compressive strength and water absorption properties of control one and PS-*b*-PAA/cement paste at the age of 28 days, according to GB/T 50081-2019 [38]. The compressive strength of each cement sample was measured three times to obtain the averaged value, and the water absorption test used water as liquid to test the water absorption of each cement paste sample. The contact angle test was used to estimate the hydrophobicity of the matrix inside the cement paste at the age of 28 days. Water contact angles were recorded by a time-elapsed camera equipped with a drop-shape analysis system (JC2000D1, Powereach, Shanghai, China). Each value reported here was obtained by averaging values measured at five different spots on the same sample. The surfaces of hardened cement paste samples were polished before the measurements. Mercury intrusion porosimetry (MIP, AutoporeV9605, Mike Instruments Co., Ltd., Hong Kong, China) and Brunauer-Emmett-Teller analysis (BET, Quadrasorb evo, Anton Paar, Graz, Austria) with 2 MP gas adsorption analyzer at 77 K was adopted to investigate the pore distribution and specific area (estimated from BJH method) of cement paste samples. For XRD, TG, and BET analysis, crushed cement paste samples were all grounded manually through an 80 μm square-mesh sieve. The calorimetric test (micro-calorimeter, TAM Air, Tbilisi, Georgia) was applied to measure the hydration heat of the cement pastes. The cement powder was added into the calculated amount of PS-*b*-PAA micelle water solution and stirred for 2 min to form a paste, and then about 15 g of paste was weighed. The hydration heat test was started 10 min after the powder was contacted with water and recorded every 10 s. During the test, both the experimental temperature and the ambient temperature were maintained at 20 °C.

### 2.2. Preparation of PS-b-PAA Micelle

The synthetic routine and schematic of the micelle are shown in Figure 1a,b, respectively. In the synthesis of PS-*b*-PAA (Figure 1a), 10.40 g of distilled styrene monomers (100 mmol), 0.28 g of 4-Cyano-4-(thiobenzoylthio)pentanoic acid (1 mmol) and 0.10 g of azobisisobutyronitrile (0.6 mmol) were added into 80 mL anhydrous toluene, and the mixed solution in a sealed flask was continuously purged with nitrogen for 30 min to remove the dissolved oxygen. Subsequently, the sealed flask was placed in an oil bath and kept at 65 °C for 24 h with magnetic stirring. Afterwards, the flask was placed in ice water to terminate the living polymerization. The solution was precipitated in methanol and repeated several times to remove any residual impurities. Finally, the pale reddish powers of PS homopolymer (9.60 g, 90% yield) were obtained. The *M*_n_ and dispersity (*Ð* = *M*_w_/*M*_n_) of the product determined by SEC were 10,300 g mol^−1^ and 1.26, respectively.

In the next step, a calculated amount of acrylic acid co-monomers (7.21 g, 100 mol) and 0.10 g of azobisisobutyronitrile (0.6 mmol) were added into the flask charged with 100 mL anhydrous tetrahydrofuran, which was dissolved with previously obtained PS homopolymer. A similar nitrogen purge was carried out before the initiation of polymerization. The sealed flask was kept at 65 °C for 24 h. Afterwards, the solution was cooled with ice water and precipitated in a mixture of methanol/water. Finally, the pale reddish viscous products of targeted diblock copolymer PS-*b*-PAA (14.45 g, 86% yield) were obtained. The *M*_n_ and dispersity (*Ð* = *M*_w_/*M*_n_) of the product determined by SEC were 17,700 g mol^−1^ and 1.23, respectively.

^1^H NMR (600 MHz, CDCl_3_, δ, ppm): 0.97–2.10 (m, protons from backbone of copolymer), 6.39–6.85 (m, o-aromatic, PS) 6.91–7.42 (m, m-, p-aromatic, PS).

The obtained PS-*b*-PAA was dissolved in a minimal amount of *N,N*-dimethylformamide and added dropwise into the water with continuous magnetic stirring. After ageing for 24 h, the obtained water solution was dialyzed in deionized water for around two weeks until no obvious signals of organic residuals in water could be detected by the total organic carbon (TOC) instrument. The concentration of PS-*b*-PAA micelle in water was further estimated by the weight analysis.

### 2.3. Preparation of Cement Pastes

The water-cement ratio (*w/c*) of PS-*b*-PAA/cement paste samples was selected, being 0.38, and the varied dosage of PS-*b*-PAA micelle in cement paste was set to 0.1, 0.2, and 0.4 wt%, respectively. The mixed proportions of all samples were concluded in Table 1. All samples were magnetically mixed for 3 min and poured into the cubic mould of 10 mm-side lengths. After being placed in an environment of 20 °C with relative humidity above 90% for 1 day, the specimens were demoulded and cured for another 28 days to obtain the hardened cement paste. The cubic hardened cement paste (10 mm×10 mm×10 mm) samples were further used for tests on water adsorption, compressive strength, and contact angle. The samples were also crushed and grounded into powers for XRD, TG, and BET analysis.

## 3. Results and Discussion

### 3.1. Characterization of PS-b-PAA and Micelle

#### 3.1.1. Characterization of PS-b-PAA

The synthesis of PS-*b*-PAA could be divided into two steps. The first step is the synthesis of homopolymer PS with an active RAFT terminator. The SEC chromatography (Appendix A) indicates a targeted molecular weight with narrow dispersity.

The following step is the synthesis of the diblock copolymer. For the targeted copolymer, PS molar fraction (*f*_PS_) is approximately controlled, being 0.50, as the previous study suggested a relatively high PAA fraction for the well-dispersed spherical micelle. The SEC chromatography also indicates a nearly constant narrow dispersity of synthesized copolymer, and the chemical composition of synthesized copolymer could be further estimated by the shift towards a higher *M*_n_ regime. The estimated composition in the diblock copolymer is highly consistent with the feeding ratio of co-monomers, which indicates the potential of precise control on molecular weight and composition of PS-*b*-PAA via RAFT polymerization. Besides, Appendix A displays the ^1^H NMR spectrum of synthesized PS-*b*-PAA, and the characteristic peaks of protons from PS segments and the backbone of the copolymer are observed. The combined evidence from SEC chromatography and ^1^H NMR spectrum indicates the successful preparation of PS-*b*-PAA with tunable molecular structure.

#### 3.1.2. Characterization of Micelle

The microstructure of PS-*b*-PAA micelle dispersed in water (Figure 2a–c) was analyzed by TEM characterizations. A spherical polymer-based particle of an averaged 80 nm diameter could be observed, which is highly consistent with the recorded particle size distribution estimated from dynamic light scattering (Appendix A). According to TEM characterizations, the self-assembled micelles were evenly dispersed following the evaporation of surrounding water molecules, and no obvious aggregation of particles was observed within the detected area.

To further investigate the microstructure of the micelle, zeta potential analysis was further applied (Figure 3). As the pH value of the water solution was gradually increased from 9 to 13, the zeta potential was accordingly changed to more negative. Since zeta potential mainly reveals the charged state of the colloid surface, it is reasonable to speculate that rich carboxylic acid groups sensitive to pH change are concentrated on particle shells. Accordingly, the hydrophobic PS-based segments as particle cores are surrounded by the hydrophilic PAA-based shells, and the speculation is consistent with the widely studied self-assembled amphiphilic BCPs in water. Besides, the negatively charged colloid surface, especially in a highly alkaline environment, may contribute to the improved dispersion stability caused by electrostatic repulsion.

#### 3.1.3. Dispersion Analysis

Figure 4a,b shows the dispersion status of PS-*b*-PAA and an equal amount of physically mixed PS and PAA homopolymers in water, respectively. The investigated polymers were both dissolved in DMF and added dropwise into the water for comparison. The observed milky appearance of the mixed polymer solution indicated the averaged particle size in micrometre scales. Meanwhile, the relatively clear water solution of PS-*b*-PAA micelle is consistent with the measured particle size smaller than 100 nm. The fact indicated an improved dispersity of PS-*b*-PAA in water compared to that of mixed two homopolymers.

### 3.2. Effects of PS-b-PAA Micelle on Rheological Properties, Compressive Strength, and Water Absorption of Cement Paste

#### 3.2.1. Rheological Properties

To evaluate the adaptability of PS-*b*-PAA micelle to the fresh cement paste, the rheological properties of the cement paste were first studied. And the rheological curves at the initial time are shown in Appendix A. It was observed that the shear stresses of the control sample and the one containing PS-*b*-PAA micelle increased linearly with the increase of the shear rates. At the initial time, the added PS-*b*-PAA micelle caused the rheological curves to increase more rapidly compared to that of the control sample without any admixture. The rapidly increased rheological curves were possibly caused by the aggregation of cement grains bridged by the negatively charged copolymers.

To investigate another important rheological parameter, the curve of shear viscosity against shear rate was also recorded, as presented in Appendix A. For both the control sample and the sample containing PS-*b*-PAA micelle, typical shear thinning behaviours of fresh cement paste at applied low and intermediate shear rates were observed. The observed shear thinning behaviours were governed by the competition between attractive colloidal interactions and hydrodynamic effects. At a low shear rate, the effect of adsorbing PS-*b*-PAA polymers on the hydrodynamic and inertia contributions to macroscopic viscosity possibly caused a shear viscosity higher than that of pristine paste. However, as the shear rate is gradually increased, the hydrodynamic effects may start to dominate all phenomena in the cement paste. Accordingly, a Newtonian plateau was observed in control and studied cement paste samples, in which shear viscosity became less dependent on shear rate. Conclusively, the rheological curves of paste cement containing PS-*b*-PAA micelle are relatively similar to those of the control sample despite the differences at low shear rates. The fact indicates that the addition of prepared PS-*b*-PAA micelle as an admixture is applicable to fresh cement.

#### 3.2.2. Compressive Strength Test

The 28-day compressive strength of 0.1, 0.2, and 0.4 wt% PS-*b*-PAA micelle-doped cement paste samples are presented in Figure 5. For a relatively low dosage of 0.1% wt, no significant decrease in compressive strength was found. As the dosage is further increased to 0.4% wt, the change in compressive strength is still not significant, compared to that of the control sample. The results demonstrated that the added PS-*b*-PAA micelles minimize the negative effects on 28-day compressive strength, which is frequently reported in cement pastes containing polystyrene-based admixture. The improvements are possibly caused by the enhanced dispersity of polymer particles with hydrophilic shells, which may effectively reduce the retarding effects of hydrophobic polystyrene on the cement hydration process without forming any blocking layer preventing water diffusion.

#### 3.2.3. Water Absorption Test

Figure 6a presents the water absorption amount variation with $\sqrt{t}$ for cement paste samples containing various amounts of PS-*b*-PAA micelles. As observed in Figure 6a, no significant linear relationship was observed in the plot of water absorption amounts against $\sqrt{t}$ for all samples. The fact suggested that the swelling characteristics of CSH caused the change in the pore structure of the cement pastes [39]. And according to the previous studies, the water absorption amount of cement paste is determined by a series of factors, including pore structure and contact angle of the liquid medium on the sample surface, as well as swelling characteristics of CSH. In this study, the discussion on the effects of swelling characteristics of CSH is omitted because of the limited experimental approaches.

The water absorption amounts of all investigated cement paste samples were rapidly increased with an elapsed time of 6 h. Among all studied samples, the control sample without any admixture exhibited the most rapid growth rate of water absorption amount, as shown in Figure 6b. After around 6 h, the water absorption amount of all samples slowly increased without any significant change as the samples are close to the adsorption saturation states. Besides, the saturated water absorption amount of cement pastes was negatively correlated with the dosage of PS-*b*-PAA micelle. The control sample exhibited the highest saturated water absorption amount, and the water absorption amount of 0.1, 0.2, and 0.4 wt% PS-*b*-PAA micelle added cement paste samples at 56 h were reduced by 7.3%, 13.7%, and 20.1%, respectively, compared to that of control one.

Besides the above analysis on water absorption amount, the widely applied water sorptivity is another gold standard in evaluating water absorption behaviour and resistance of cement paste to water absorption. Herein the evolution of water sorptivity and relative water sorptivity change with water absorption amount for investigated cement paste samples are shown in Figure 6c,d, respectively. At any given water absorption amount, the water sorptivity of the control sample is higher than the three doped ones (Figure 6c), and the result was highly consistent with the recorded water absorption amount of all samples. Figure 6c shows that the change of water sorptivity with the increased water absorption amount, and the curve shapes of three doped samples were all similar to those of the control one. Taking the control sample as a reference, the relative water sorptivity of doped samples all exhibited convex curves plotted with water absorption amount (Figure 6d), and the relative water sorptivity at saturated water absorption amount was negatively correlated with the dosage of PS-*b*-PAA micelle, indicating that the added polymer micelles indeed change the swelling characteristics of CSH and pore structure of the cement paste.

### 3.3. Effects of PS-b-PAA Micelle on Hydrophobicity, Hydration Process, and Pore Structure of Cement Paste

#### 3.3.1. Contact Angle Test

The time-dependent contact angles of surface polished cement paste samples are presented in Figure 7. The measured initial contact angle of samples was in the following order: 0.4% wt doped one > 0.2% wt doped one > 0.1% wt doped one > control sample. The fact indicates the gradually enhanced hydrophobicity of the cement paste matrix caused by the increased incorporation of PS-*b*-PAA micelle. The enhanced hydrophobicity of the matrix obviously led to the reduced water adsorption amount of cement paste. The recorded time for macroscopic water droplet absorption followed an order similar to the above. With elapsed time, the measured contact angles of samples were all gradually decreased. And the time for macroscopic water droplet absorption of 0.4 wt% doped sample was postponed to over 10 min, compared to 7 min of the control sample. In combination with the measured initial contact angle and the time taken for water absorption, the hydrophobicity and water resistance of cement paste were found to increase by the doped PS-*b*-PAA micelle.

#### 3.3.2. Hydration Process Analysis

To quantitatively estimate the CH content of hydration products, TGA characterization is carryout out, and TG curves of samples are presented in Figure 8a. Several separated weight loss stages could be clearly observed as follows: the first stage, between 50–200 °C, corresponding to the loss of evaporable water, dehydration of C–S–H gel and ettringite, the second stage between 400–500 °C is mainly caused by the dehydration of CH, and the last stage between 500–800 °C is related to the decarbonization of CaCO_3_. Based on the estimation of the second stage, the CH content of the control and 0.4% wt doped cement paste samples at 7 days was 22.51% and 23.64%, respectively. At 28 days, the CH contents increased to 23.96% and 24.81%, respectively. Obviously, the incorporation of admixtures into the cement paste imposed no significant effects on CH content during 7–28 days of hydration, which serves as an indicator of hydration degree.

To further determine the degree of hydration, XRD spectra of cement paste samples were collected and shown in Figure 8b. Commonly observed characteristic peaks of CH, C_3_S, and C_2_S could be discovered in studied samples. The degree of hydration was revealed by the main diffraction peaks of CH (18.04° and 34.20°) and C_3_S/C_2_S (32.20°). In comparison with those of the control sample, the intensities of main diffraction peaks in spectra of 0.4% wt doped cement paste sample are quite similar. Following 28 days of curing, although the diffraction peaks of CH generally became more intensive, no obvious difference between the control and 0.4% wt doped ones could be observed. The observation suggested the minor effects of doped PS-*b*-PAA micelle on the hydration process of cement clinker, which was also consistent with the results obtained from TGA analysis.

Besides TGA and XRD analysis, the exothermic heat flow curve of investigated cement pastes, as shown in Figure 8c, was also collected for the evaluation of the hydration process. The cumulative released heat from the exothermic reaction indicates the degree of the cement hydration process, which is closely related to the development of microstructure and macroscopic strength. Compared to the heat flow curve of the control sample, the addition of PS-*b*-PAA micelle caused the inhibiting effects with reduced maximum heat flow, as well as the retarding effects with delayed occurred flow peak. As the dosage of PS-*b*-PAA micelle was increased, the inhibiting and retarding effects became more intensive within one day after the hydration occurred. However, the difference in accumulative heat in 3 days was not obvious, as the accumulative heat of 0.4 wt% doped sample was only reduced by 8.8%, compared to that of the control one. Therefore, it was worth noting that the effects on the hydration process from added PS-*b*-PAA micelle were limited in this study, despite the observed inhibiting and retarding effects within one day.

Conclusively, the results of isothermal microcalorimetry, XRD, and TG were highly consistent with each other. The hydration degree of investigated cement paste samples was less affected by the added PS-*b*-PAA micelle, and thus the 28-day compressive strength of doped cement paste could be kept without significant loss.

#### 3.3.3. Pore Structure Analysis

Besides the hydrophobicity of the matrix, another factor governing the water resistance of paste cement is the pore structure inside the matrix. The pore size distributions of the control sample and 0.4 wt% doped one were characterized by MIP and BET, respectively. As presented in Figure 9 and Table 2, the cumulative pore volume of the control sample and 0.4 wt% doped one were relatively close, and the cumulative pore volume of 0.4 wt% doped sample was slightly lower than that of the control one, which is in response to the porosity of 23.2% slightly lower than 23.5% of control one. Besides, the estimated total pore area and median pore diameter of compared two samples were all similar, indicating an insignificant change in pore structure with the incorporation of PS-*b*-PAA micelle. According to the pore size distribution of cement paste, the pore size is generally categorized into the following four scopes: gel pores (3–10 nm), small capillary pores (10–100 nm), big capillary pores (100 nm–1 μm) and macro pores (>1 μm). Since MIP characterization results revealed no obvious difference in capillary pores’ structure, the potential change in the smallest gel pores became of great importance. Therefore, BET characterization was further carried out to reveal the gel pore structure. As shown in Appendix A, the estimated cumulative surface area of pores and volume of pores of 0.4 wt% doped sample within the tested scope were both 20% larger than those of the control one, which was possibly caused by the added averaged 80 nm-sized micelles with a high specific area. Thus, an increased pore volume was observed for pores larger than approximately 30 nm. Besides, the estimated average pore diameters of two compared samples corresponding to the observed peaks in Appendix A were found both consistent with those estimated from MIP. In comparison with the control sample, the unchanged porosity and pore structure in the polymer doped sample indicated the unaffected hydration process of cement clinker, which further caused the insignificant compressive strength loss at 28 days.

Based on the above comprehensive analysis of properties of cement paste, the main causes for the improved water resistance without any sacrificed compressive strength were concluded as follows:Improved dispersity of hydrophobic polystyrene. When nanometer-sized PS-*b*-PAA micelles are formed in the aqueous phase, an amphiphilic core-shell structure with hydrophobic PS segments surrounded by hydrophilic PAA is assembled. The surface of well-dispersed nanoparticles with rich charged carboxylic groups may avoid the commonly occurring aggregation of PS nanoparticles via the electrostatic repulsion.Hydrophobic modification of the matrix. The added PS-*b*-PAA micelle significantly improves the hydrophobicity of the cement paste matrix compared with that of the control one. The carboxylic groups on micelle surfaces are possibly complex with the calcium ion from the hydration products, and the inner wall of the pore structure is thus covered by the hydrophobic PS segments. The bulk hydrophobic modification caused by PS segments further leads to the reduced water absorption of the cement paste matrix.Unaffected hydration process and pore structure of cement paste. Although PAA segments in micelle may retard and inhibit the hydration process of cement clinker at an early stage, the adverse effects on hydration are considered less than those of polystyrene emulsion. For the control and doped samples, the hydration products and pore structure at late hydration time (7–28 days) are similar, and thus the compressive strength of cement paste is less affected by the added PS-*b*-PAA micelle.

## 4. Conclusions

In this study, an amphiphilic diblock polystyrene-*block*-poly(acrylic acid) (PS-*b*-PAA) was synthesized via RAFT polymerization, which enabled the potential of precise control on the chemical compositions and molecular weights of synthesized copolymers. The prepared PS-*b*-PAA self-assembled into an averaged 80-nm sized micelle with hydrophobic core/hydrophilic shell nanostructure in water, which was characterized by TEM, DLS, and zeta potential tests. The effects of micelles as admixture on cement paste were further investigated. The reduced water absorption, as well as unaffected compressive strength observed in macroscopic, were deeply investigated by the contact angle, calorimetry, XRD, TGA, MIP, BET, and other characterizations. The specific conclusions are drawn as follows:In the water phase, the hydrophilic core consisted of PS segments surrounded by the hydrophilic shell of PAA segments. The hydrophilic shell surface, along with the negatively charged carboxylic groups on the surface, may improve the dispersity of nanometer-sized PS particles, especially in a highly alkaline environment, compared to that of emulsified PS particles with micrometre size.Compared with the control sample, the cement paste exhibited similar rheological behaviours, gradually reducing the water absorption amount and mostly unaffected compressive strength, as the dosage of PS-*b*-PAA micelle was increased from 0.1 wt% to 0.4 wt%. For a dosage of 0.4 wt%, the water absorption amount of cement paste at 56 h was reduced by 20.1% in comparison with the control sample.The main contributor to enhanced water resistance was discovered to be the enhanced hydrophobicity of cement paste caused by the added copolymers, which were indicated by the measurements on initial contact angle and the time of macroscopic water absorption. Besides, the effects of added copolymers on the hydration process and pore structure of cement paste were not obvious, especially after 3–28 days of curing. The less affected hydration degree and pore structure reasonably caused the mostly unchanged 28-day compressive strength of cement paste.

In summary, the advantages and application potential of amphiphilic PS-*b*-PAA micelle, as prospective substitutes for emulsified PS particles commonly used in cement/concrete, were demonstrated in this study. The water absorption resistance of cement paste could be effectively enhanced by a relatively low dosage of prepared PS-*b*-PAA micelle. Meanwhile, the strength loss caused by traditional PS emulsion was prevented. Therefore, the massive deterioration problems associated with water transportation, such as sulfate attack, chloride invasion, and freeze-thaw damage, could be addressed, and the service life of cement-based materials in harshly humid environments could be effectively extended with construction and cost convenience.

## Figures and Tables

**Figure 1 materials-16-02190-f001:**
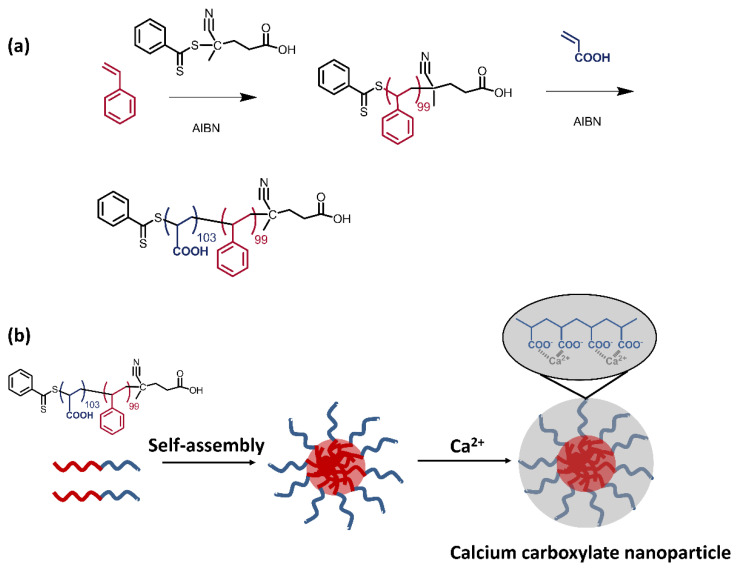
(**a**) Synthetic routine of targeted PS-*b*-PAA and (**b**) schematic of amphiphilic micelle structure from self-assembly in water solution of cement paste.

**Figure 2 materials-16-02190-f002:**
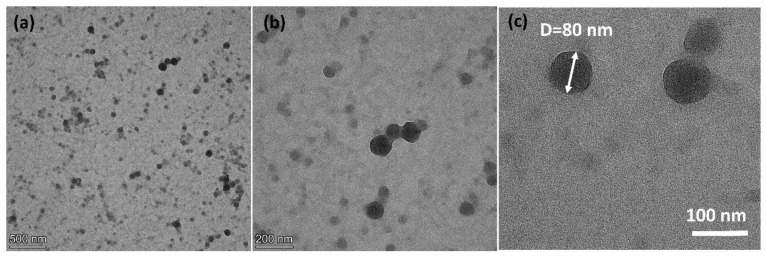
(**a**,**b**) Transmission electron microscopy images of dispersed PS-*b*-PAA micelle following the evaporation of water in different scales and (**c**) the averaged diameter of observed micelles is estimated to be around 80 nm.

**Figure 3 materials-16-02190-f003:**
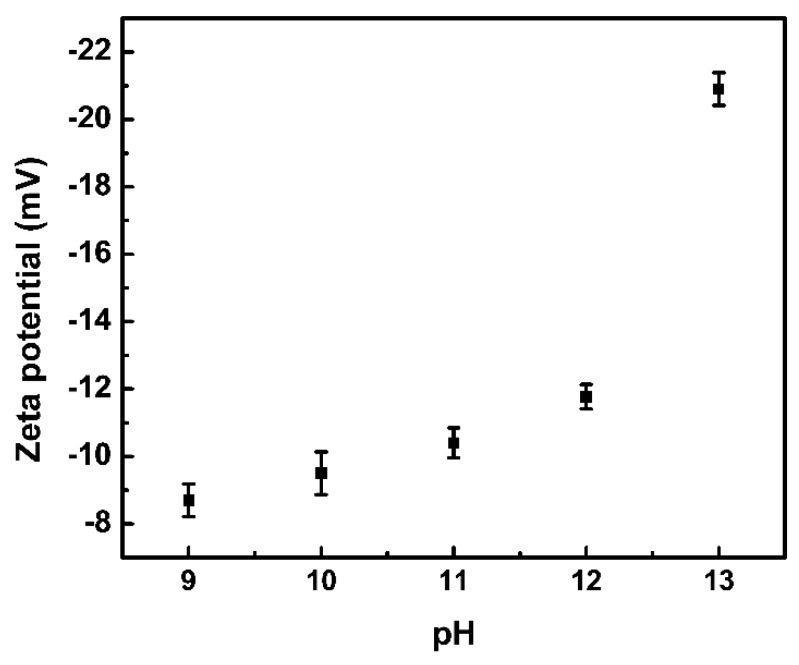
Zeta potential of PS-*b*-PAA micelle in water under various pH conditions with error bars.

**Figure 4 materials-16-02190-f004:**
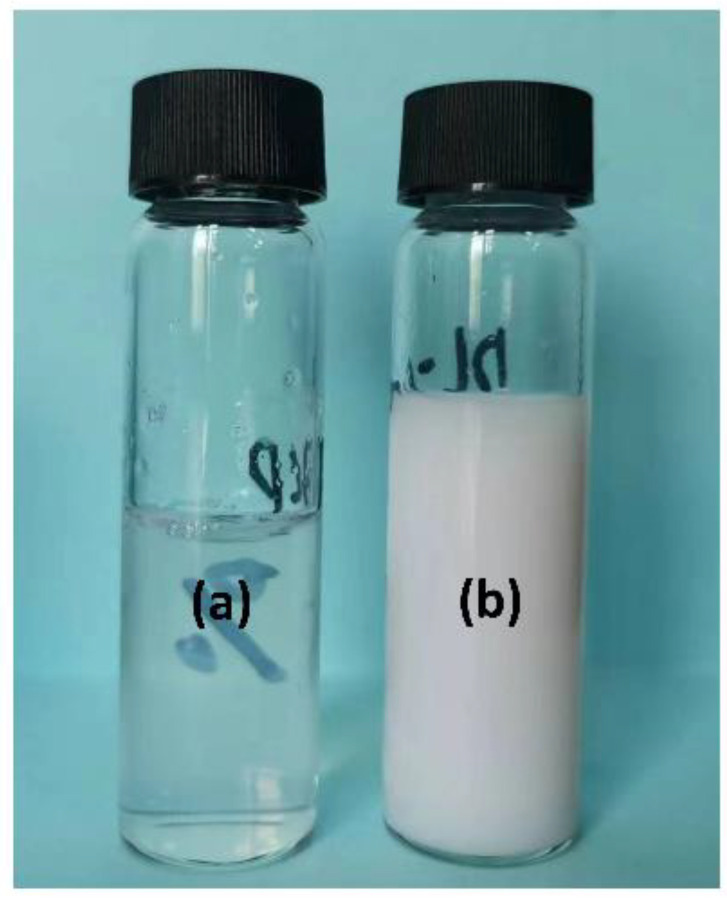
The dispersion status of (**a**) PS-*b*-PAA and (**b**) an equal amount of physically mixed PS and PAA homopolymers in water. And all polymers were pre-dissolved in DMF solvent and added dropwise into water.

**Figure 5 materials-16-02190-f005:**
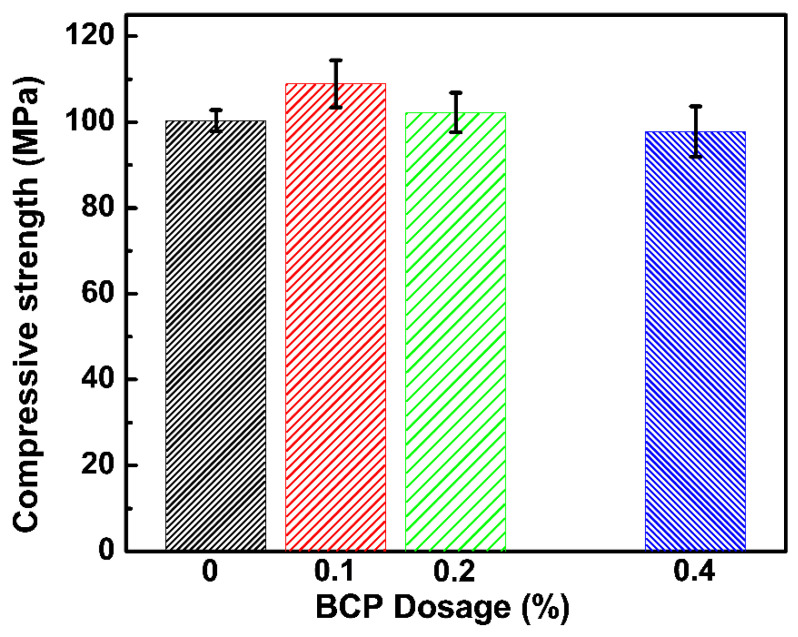
28-day compressive strength of 0.1, 0.2, and 0.4 wt% PS-*b*-PAA micelle doped cement paste samples. The control one is also presented for comparison.

**Figure 6 materials-16-02190-f006:**
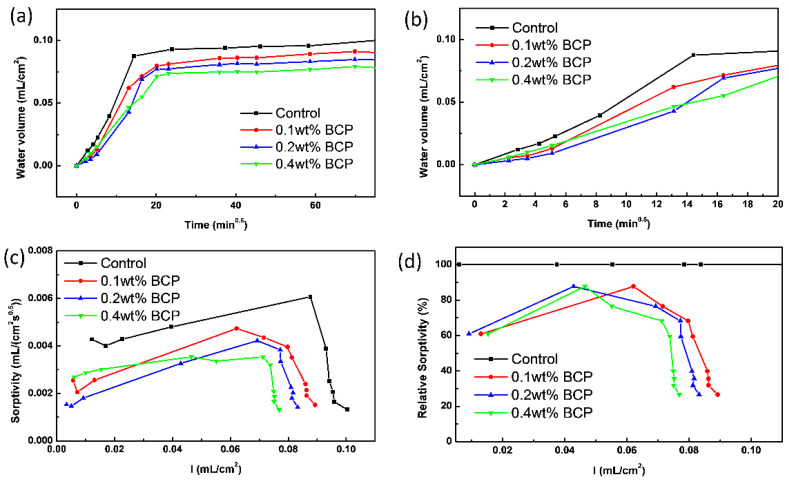
Water absorption amount variation with $\sqrt{t}$ for control and BCP doped cement pastes samples (**a**) at full time and (**b**) at an early stage, (**c**) evolution of water sorptivity, and (**d**) relative water sorptivity variation with water absorption amount for control and BCP doped cement pastes samples.

**Figure 7 materials-16-02190-f007:**
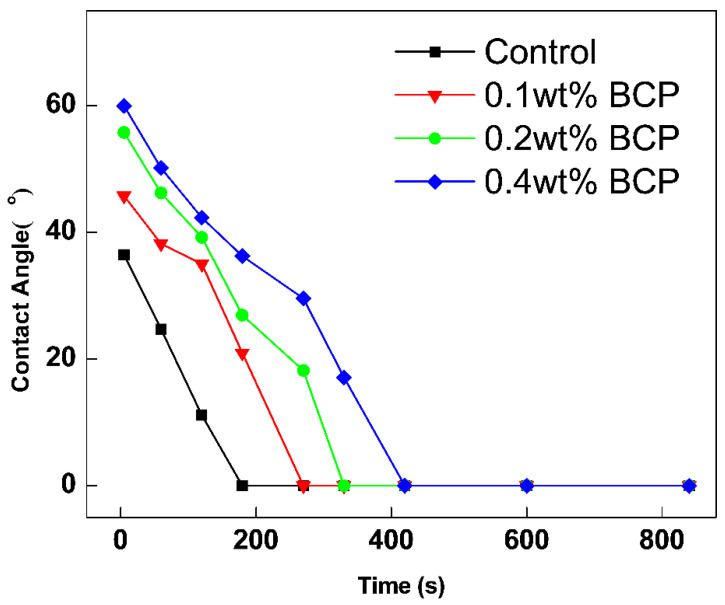
The variation of water contact angle with elapsed time for polished control and BCP doped cement pastes samples.

**Figure 8 materials-16-02190-f008:**
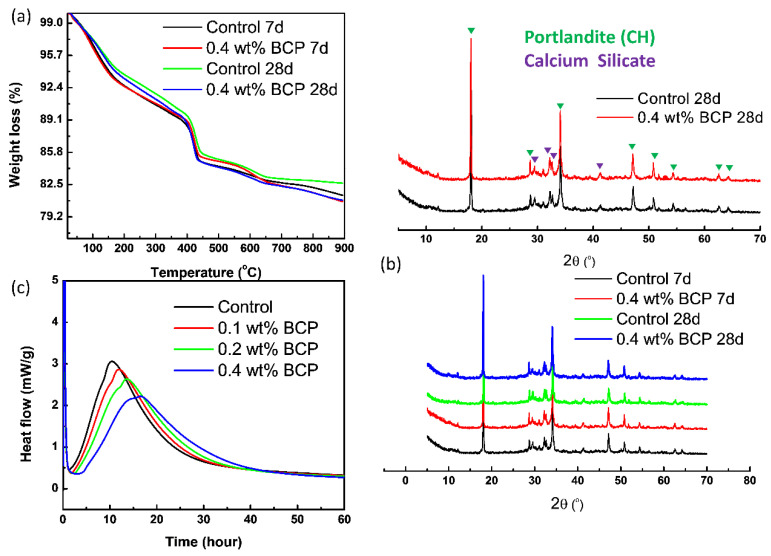
(**a**) TGA curves, (**b**) X-ray diffraction spectrums, and (**c**) exothermic curve of control and BCP doped cement pastes samples.

**Figure 9 materials-16-02190-f009:**
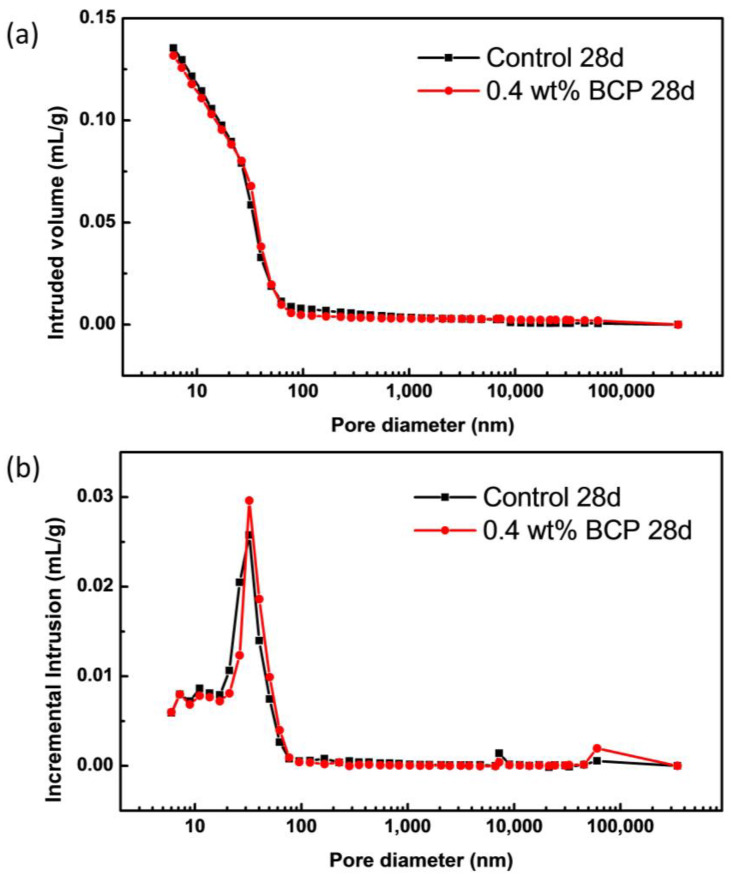
(**a**) Cumulative pore volume and (**b**) incremental pore volume of control and BCP doped cement pastes samples collected from mercury intrusion porosimetry.

**Table 1 materials-16-02190-t001:** Mix proportions of various PS-*b*-PAA (BCP) doped cement paste samples (mass in a unit of g).

Sample	Cement (g)	Water (g)	PS-*b*-PAA Micelle (g)
Control	200	76	/
0.1 wt% BCP	200	76	0.276
0.2 wt% BCP	200	76	0.552
0.4 wt% BCP	200	76	1.104

**Table 2 materials-16-02190-t002:** Basic parameters of pore structure estimated from MIP characterization.

Sample	Total Intrusion Volume (mL/g)	Total Pore Area (cm^3^/g)	Median Pore Diameter (nm)	Porosity (%)
Control	0.1355	26.44	29.85	23.49
0.4 wt% BCP	0.1318	25.32	33.00	23.20

## Data Availability

Datas can be found in the Appendix A.

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
