# Peer review of "Effects of an Amphiphilic Micelle of Diblock Copolymer on Water Adsorption of Cement Paste"

_materials, 2023, doi:10.3390/ma16062190_

Round 1

Reviewer 1 Report

The article under review is well written, structured, contains a fairly large number of studies.

There are some comments on the article:

1) the introduction does not provide any information about the effect of the amphiphilic micelle of the diblock copolymer on the properties of the cement paste;

2) the research methods do not indicate on which devices the shear rate and shear stress of the cement paste were studied;

3) why is the degree of hydration of portland cement determined only by comparing the peaks of portlandite and calcium hydrosilicate? no source of information specified, or method

4) there is no method for determining the contact angle of wetting

5) there is a lack of conclusion about the optimal dosage of the studied additive.

Reviewer 2 Report

The article "Effects of an amphilic micelle of diblock copolymer on water adsorption of cement paste" is interesting and valuable. However, it contains some inaccuracies that should be addressed.

The work is careless in terms of editing. It contains a lot of typing errors, such as a lack of spaces or capital letters, etc.

The introduction should be extended to information and citations about cement modifications with polymer particles, with their weight ratios.

The experimental part should be supplemented with:

• Description of Dynamic Light Scattering (including apparatus model and manufacturer as well as research procedure with sample preparation).

• Description of Contact Angle Tests (including apparatus model and manufacturer as well as research procedure with sample preparation).

• Description of XRD (including apparatus model and manufacturer as well as research procedure with sample preparation).

• Description of the compressive tests and water absorption, taking into account the dimensions, shape, and method of preparation of the samples. Information on whether these properties were examined for hardened pastes or unpaved pastes should be given. This is not clearly stated. In the 190-192 lines, the authors write about curing: " After being palced in the environment of 20 oC and the relative humidity above 90% for 1 days, the specimens were demoulded and cured for another 28 days to obtain the hardened cement paste." In my view, hardened cement should not be named as a paste. The paste by the name indicates a liquid. So the authors’ term “paste cement” should be reviewed.

• Perhaps there is a lack of DSC description (depending on what technique was used to receive "Exothermic Curves" in Figure 8).

Line 340 - The authors use the term "enhanced hydrophobicity". It is decreased hydrophilicity because the Water Contact Angle indicates that the tested surface is still hydrophilic.

Figure 8:

• The term "thermal gravity curves" is incorrect

• XRD - Was It Powder Diffraction? If so, complement the experimental part.

• How the "Exothermic Curves" Were Obtained? - Was it DSC? If it was DSC it should be described.

• The origin of the Exothermic Effect should be explained in the discussion.

Round 2

Reviewer 2 Report

Accept in present form